# Awareness and Perceptions among Members of a Japanese Cancer Patient Advocacy Group Concerning the Financial Relationships between the Pharmaceutical Industry and Physicians

**DOI:** 10.3390/ijerph19063478

**Published:** 2022-03-15

**Authors:** Anju Murayama, Yuki Senoo, Kayo Harada, Yasuhiro Kotera, Hiroaki Saito, Toyoaki Sawano, Yosuke Suzuki, Tetsuya Tanimoto, Akihiko Ozaki

**Affiliations:** 1Medical Governance Research Institute, Minato-ku, Tokyo 108-0074, Japan; senooyuki0821@gmail.com (Y.S.); kayoharada.0615@gmail.com (K.H.); yousuzuki_mail@yahoo.co.jp (Y.S.); tetanimot@yahoo.co.jp (T.T.); ozakiakihiko@gmail.com (A.O.); 2School of Medicine, Tohoku University, Sendai 980-8574, Japan; 3School of Health Sciences, University of Nottingham, Nottingham NG7 2RD, UK; y.kotera@derby.ac.uk; 4Department of Gastroenterology, Sendai Kousei Hospital, Sendai 980-0873, Japan; h.saito0515@gmail.com; 5Department of Surgery, Jyoban Hospital of Tokiwa Foundation, Iwaki 972-8322, Japan; toyoakisawano@gmail.com; 6Department of Internal Medicine, Navitas Clinic Kawasaki, Tokyo 210-0007, Japan; 7Department of Breast and Thyroid Surgery, Jyoban Hospital of Tokiwa Foundation, Iwaki 972-8322, Japan

**Keywords:** conflict of interest, ethics, Japan, financial relationship, patient-centred care, pharmaceutical industry

## Abstract

Objectives: Awareness and perceptions of financial conflicts of interest (FCOI) between pharmaceutical companies (Pharma) and healthcare domains remain unclear in Japanese cancer patient communities. This study aimed to assess awareness (RQ1), the influence of FCOI on physician trustworthiness (RQ2), and their perception (RQ3) among the Japanese cancer patient advocacy group members. Methods: A cross-sectional study using a self-administered survey was conducted with a Japanese cancer patient advocacy group between January and February 2019. The main outcome measures included awareness and perceptions of physician–Pharma interactions, their impact on physician trustworthiness, and attitudes towards FCOI among medical and other professions. Furthermore, we performed thematic analyses on the comments which responders provided in the surveys. Results: Among the 524 contacted members, 96 (18.3%) completed the questionnaire, including 69 (77.5%) cancer patients. In RQ1, most of the respondents were aware of physician–Pharma interactions, although the extent differed based on the nature of the interaction. Furthermore, the respondents mainly considered these interactions influential on clinical practice (RQ2) and agreed to the need for further regulation of physician–Pharma interactions (QR3). In qualitative analyses (*n* = 56), we identified the 4 following themes: perception towards the FCOI (Theme 1), concerns about the respondent’s treatment (Theme 2), reason of physician–Pharma interactions (Theme 3), and possible solutions from the patient perspective (Theme 4). Conclusions: Most respondents were generally aware of physician–Pharma-associated FCOI and perceived them negatively. Additionally, participants appeared supportive of further FCOI regulation to protect patient-centred care. Abbreviations: FCOI—financial conflicts of interest; United States—US; Pharma—pharmaceutical companies; RQ—research question.

## 1. Introduction

In recent years, patient-centred care has widely been considered a key component of healthcare delivery. Defined as one of the six domains of healthcare quality, introduced by the National Academy of Medicine [1], patient-centred care focuses on delivering healthcare which respects patient preferences, needs, and values. Therefore, financial conflicts of interest (FCOI) between pharmaceutical companies (Pharma) and healthcare domains are of significant concern, mainly due to potential bias in patient care in the operation of medical institutions [2,3,4,5,6]. Key criteria of patient-centred care include full transparency and fast delivery of information [7,8,9]. Globally, there are transparency initiatives such as the Sunshine Act and Open Payments Database in the United States (US) [10,11,12], Transparency in Healthcare in France [5], Euro for Doctors and LeitlinienWatch in Germany [13,14], Disclosure UK in the United Kingdom [15,16,17], Disclosure Australia [18,19], and the Money Database in Japan [2,20,21]. These initiatives all emphasize improving transparency and raising awareness of FCOI among patients and the general public to enhance patient-centred care.

However, there remains an ongoing controversy about whether these transparency initiatives improve the general awareness of FCOI [10,12,22]. A previous qualitative study conducted before these initiatives reported that most cancer clinical trial respondents had a positive or neutral view of physicians receiving research funding from Pharma [23]. Moreover, there has been little improvement in awareness of FCOI even after the launch of transparency initiatives [10,22,24]. Rather, it has been reported that public trust in healthcare professionals dropped following the launch of these initiatives [25]. These findings demonstrate the limited effect on the awareness of FCOI by these transparency initiatives.

Nonetheless, recent discussions on this issue have not considered specific populations with a higher interest in this topic, including patients with critical illnesses or their caregivers. Among patient populations, cancer patients are particularly important, given the critical nature of the disease and its burden [26], as well as the development of numerous novel and expensive therapeutics [27,28]. Since novel anticancer drugs are potentially highly profitable, these agents’ development remains among Pharma’s highest priorities [29]. Consequently, it remains crucial for cancer patients to understand FCOI among Pharma and healthcare domains.

The issues surrounding FCOI are relevant in Japan due to its universal health coverage and the fact that its pharmaceutical market is the third largest globally, after the US and China [30]. In 2018, anticancer drugs’ annual pharmaceutical sales exceeded JPY 1.24 trillion (USD 11 billion), accounting for 12% of Japan’s total annual pharmaceutical sales. Despite an overall decline in Japan’s pharmaceutical sales, the oncology drug market continues to expand and is projected to exceed USD 13 billion by 2025 [31]. There also has been progress in terms of Japanese transparency initiatives. Along with several non-profit organizations, the Japan Pharmaceutical Manufacturer Association has, since it first developed transparency guidelines in 2011 [32], led efforts to improve transparency in Pharma and healthcare domains’ financial relationships. For instance, since 2013, all Japan Pharmaceutical Manufacturer Association members have voluntarily disclosed payments to physicians on their respective websites following established transparency guidelines [32]. Additionally, Japanese non-profit organizations, including the Medical Governance Research Institute and Tansa (formerly known as Waseda Chronicle), have developed the Money Database. This database enables the general public to learn about financial relationships among individual pharmaceutical companies and healthcare sectors [2,33,34,35,36]. While these organizations disclose industry payments made to physicians, the question of how Japanese cancer patients perceive the information remains unclear. Accordingly, the present study aimed to examine awareness of physician–Pharma interactions and their impact on physicians’ trust, while appraising perceptions of these payments among Japanese cancer patients and cancer patient advocates.

### Study Aims

This study aimed to assess the awareness and perception of FCOI between physicians and Pharma among cancer patients and to explore its influence on patients’ trust towards physicians and care. Furthermore, this study intended to discuss possible clinical implementation and improvements in the transparency of the FCOI between Pharma and healthcare sectors in Japan. We established the following three research questions to navigate the study:

RQ1: Awareness—How familiar are Japanese cancer patients with the physician–Pharma interactions?

RQ2: Influence—By what factors do the physician–Pharma interactions influence patients’ trust and care?

RQ3: Perception—What are Japanese cancer patients’ perceptions of physician–Pharma interactions?

## 2. Methods

### 2.1. Study Setting, Design, and Respondents

This study was performed in cooperation with the Cancer Treatment Society for Citizens, a support group for cancer patients established in 2004. In response to increasing trends towards second opinions in oncology [37], this support group aids patients in determining the most appropriate treatment by offering referrals chiefly with radiation oncologists certified by both the Japan Radiology Society and the Japanese Society for Radiation Oncology. Along with the referral service for second opinions, the Cancer Treatment Society for Citizens also offers educational programs and publishes articles by cancer specialists on its website to promote general awareness of contemporary cancer research and treatment options. Therefore, the target population of this study includes cancer patients and their caregivers.

### 2.2. Data Collection

Data collection occurred from 9 January to 10 February 2019, using a structured questionnaire. The questionnaire was distributed to all 524 members registered on the mailing list of Cancer Treatment Society for Citizens, with the society’s quarterly newsletter on 9 January 2019. Members that agreed to participate in the study completed the questionnaire and returned it via the enclosed self-addressed, stamped envelope by 10 February 2019. To mitigate response bias, the authors were not directly involved in the survey distribution or collection.

### 2.3. Survey Sheet

We compiled our cross-sectional questionnaire survey, taking into account previous studies and the local context of Japanese cancer care [38,39]. The survey included 51 questions covering the following items: (1) the status of disease progression and demographic characteristics, such as gender, age, education level, and medical history (12 items); (2) awareness of physician–Pharma interactions, including physician receipt of gifts, payments, and research rewards from Pharma (11 items; e.g., *“Do you know that some physicians would receive pamphlets or leaflets concerning products manufactured by pharmaceutical companies from their sales representatives?”*) on a 3-point Likert scale (*Yes, No, or Not sure*: α = 0.82); (3) influence of the interactions on trust in physicians (11 items; e.g., *“How would your trust in your physician be influenced if your physician received pamphlets or leaflets with information about the products from a pharmaceutical sales representative?”*) on a 5-point Likert scale (α = 0.89); (4) perception of trust and regulation on the interactions (11 items; e.g., *“Stricter regulations about gifts, meals, and honoraria from pharmaceutical companies to physicians are needed”.*) on a 5-point Likert scale (α = 0.72); (5) attitude towards the industrial payments across professions (5 items; e.g., *“It is problematic for physicians to receive gifts, meals, and other entertainment from pharmaceutical sales representatives”.*) on a 5-point Likert scale (α = 0.65); and (6) an open-ended question about respondents’ perception on the FCOI between Pharma and physicians (“*Please freely describe your thoughts about non-research-related offerings from Pharma to a physician (e.g., gifts, free meals, monetary incentives for a lecture)*”). The survey questions are described in greater detail in the Appendix A. Validity and reliability of all variables assessed using a Likert scale were deemed acceptable–high: validity was confirmed by an expert panel and data distribution, and reliability was confirmed using Cronbach’s alpha 0.60 ≤ α ≤ 0.95 [40].

### 2.4. Data Analysis

The initial descriptive analysis included the survey respondents’ sociodemographic and clinical variables and all questions concerning respondent awareness, influence on trust in physicians, perceptions, and attitudes towards FCOI.

Next, to assess the association between awareness (outcome variable 1) and each respondent’s sociodemographic and clinical factors, we first evaluated the respondents’ factors stratified by this outcome variable. We recategorized the following variables as appropriate: annual income, highest educational qualification, medical history, type of business, hospital types, cancer stage, year of diagnosis, experience with cancer recurrence, experience with pharmacotherapy, experience with radiotherapy, and experience with surgical treatments. Similarly, with regard to the awareness, to ensure statistical stability, the awareness status was recategorized into two groups, with the outcome variable set as those who responded “*Yes*” to at least one of the 11 questions about awareness, and those who responded with *”No”* or *“Not sure”* to all questions.

We then constructed the logistic regression models for awareness to evaluate the relationship between this outcome variable and respondent’s sociodemographic and clinical factors. Regrouping of the outcome variable and covariates (sociodemographic and clinical factors) were conducted in the same manner as the stratified analysis above. An assessment of multicollinearity was conducted with the cutoff value of variation inflation factors (VIFs) <10.

Furthermore, we similarly conducted the stratified analysis and logistic regression analysis to evaluate the respondents’ factors associated with the impact of physician–Pharma interaction on trust in physicians (outcome variable 2). In this analysis, the questions measuring how physician–Pharma interactions would affect trust in physicians were aggregated into two types, with the outcome variable set as those who reported “*decreased trust in physicians*” or “*slightly decreased trust in physicians*” on at least one question, and the other respondents. Regrouping of the covariates (sociodemographic and clinical factors) and assessment of multicollinearity were conducted in the same manner as the analysis for awareness.

When compared with Section 3.2.1, which investigated the awareness on physician–Pharma interaction, and Section 3.2.2, which investigated the impact of physician–Pharma interaction on trust in physicians, Section 3.2.3 and Section 3.2.4 incorporated various kinds of items, and it was difficult to set one factor to be narrowed down as the outcome variable to be explored in the regression analyses in these sections.

These data analyses basically followed similar previous studies reported by Ammous et al. and Green et al. [38,41]. All analyses, including descriptive statistics, were performed using Stata version 15 (STATA Corp., College Station, TX, United States). Conversion of JPY to USD used the 2019 average monthly exchange rate of JPY 109.0 per USD 1.

Lastly, respondents’ responses to the open-ended questions on perceptions of FCOI were analysed thematically, following Braun and Clarke [42]. This consisted of the following five steps: (1) familiarisation of the data, (2) generating initial codes, (3) searching for themes, (4) reviewing themes, and (5) defining themes. First, one member (A.M.) of our research team repeatedly read all the open-ended responses, identifying the units of meaning and generating codes to capture key thoughts and concepts contained in the responses. Next, this same team member categorized the codes thematically to identify themes. Verification of these themes, including their coherence and distinctiveness, was executed by the entire research team. This collaborative effort helped mitigate any unperceived biases associated with our individualized interests and prior research experience on financial and nonfinancial COI among healthcare professionals in Japan and the US.

### 2.5. Ethics Approval

The Institutional Review Board of Medical Governance Research Institute granted ethics approval of this study (MG2018-07-0928), adhering to guidelines established by the Japanese Ministries of Health Labour and Welfare, and Education, Culture, Sports, Science, and Technology.

## 3. Results

### 3.1. Respondents

All surveys returned by 10 February 2019 were considered. Of the 524 eligible survey respondents, 96 completed the questionnaire (completion rate = 18.3%). Table 1 presents respondent sociodemographic and clinical characteristics. Respondents were predominantly male (67.7%, 63/93), older than 70 years (52.3%, 49/94), and 53.8% (50/93) had educational attainment of a bachelor’s degree or higher. Furthermore, 55.9% (52/93) were unemployed, and about half (46.5%, 40) had an annual household income of over 36,697 USD, roughly the average of Japanese households in 2018. Of the 89 respondents providing primary disease status, 69 (77.5%) were cancer patients, and 20 (22.5%) were non-cancer patients (e.g., family members of cancer patients).

Of the 69 respondents self-identifying as cancer patients, 75.4% (52/64) received their diagnosis before 2015. Furthermore, 30.9% (21/68) of these respondents primarily received treatment at a municipal hospital, while 60.9% (42/69) did not actively receive any treatment during the survey period. Lastly, 39 (56.5%) reported previous pharmaceutical treatments for cancer, including anticancer drugs, molecularly targeted drugs, or hormone therapy.

### 3.2. Quantitative Findings

#### 3.2.1. Awareness of Physician–Pharma Interactions

Figure 1 presents the respondents’ awareness of physician–Pharma interactions. The proportion of the respondents aware of these interactions ranged from 2.1% to 65.3%, depending on the type of interaction. The interaction with the largest proportion of respondent awareness included pamphlets and leaflets (65.3%), followed by stationery (64.2%), and free drug samples (53.1%). In contrast, the respondents were least aware of stock ownership (2.1%). Although Japanese companies belonging to the Japan Pharmaceutical Manufacturer Association have disclosed payments to healthcare sectors since 2013, only 10.5% (10/95) of survey respondents were aware of such disclosures. Overall, 80.2% (77/96) of respondents were aware of at least 1 physician–Pharma interaction.

We illustrated the patients’ characteristics stratified by awareness on physician–Pharma interactions in Table 2 and findings of logistic regression models for awareness on physician–Pharma interaction in Table 3. None of the sociodemographic and clinical variables in the logistic regression analysis were significantly associated with awareness of physician–Pharma interactions.

#### 3.2.2. Influence of Physician–Pharma Interactions on Trust in Physicians

Figure 2 shows the impact of physician–Pharma interactions on respondents’ trust in physicians. Although respondents overwhelmingly responded that most physician–Pharma interactions neither increased nor decreased their trust in physicians, 81.2% (58.3% decrease in trust, 22.9% slightly decreased trust) agreed that stock ownership would negatively impact their trust in physicians. Additionally, accepting honoraria for registering patients in industry-sponsored research (52.1% decrease in trust, 26.0% slightly decreased trust) or lecture fees (31.6% decrease in trust, 30.5% slightly decreased trust) also substantially impacted respondent perceptions. Interestingly, few respondents reported that indirect gifts, such as accepting free samples of prescription drugs (12.8% decrease in trust, 25.5% slightly decreased trust), would reduce trust in physicians. On the other hand, participation in industry-sponsored research generally appeared to increase physician trustfulness by 13.6% (increased trust 2.1%, slightly increased trust 11.5%) among respondents. Overall, 90.6% of respondents marked decreased trust or slightly decreased trust in at least 1 question.

Again, we have illustrated the patients’ characteristics stratified by the impact of physician–Pharma interaction in Table 4 and the findings of logistic regression models for the impact of physician–Pharma interaction in Table 5. None of the sociodemographic and clinical variables in the logistic regression analysis were significantly associated with awareness of physician–Pharma interactions.

#### 3.2.3. Perception on Physician–Pharma Interactions

Figure 3 details the reported perceptions on a series of statements regarding physician–Pharma interactions and regulations associated with each. Overall, a substantial proportion of respondents either agreed or slightly agreed that gifts from pharmaceutical companies have a significant influence on physicians’ prescription behaviour (35.8% agree, 38.9% slightly agree). A similar proportion viewed such gifts as unethical (31.6% agree, 30.5% slightly agree), contributing to unnecessary prescriptions (34.7% agree, 37.9% slightly agree), and negatively influencing respondents’ trust in physicians (38.9% agree, 31.6% slightly agree). Additionally, while many respondents acknowledged the need to regulate physician–Pharma interactions, more concluded that there should be greater self-regulation by industry (60.0% agree, 26.3% slightly agree) or physicians (60.2% agree 24.7% slightly agree), as opposed to legal regulation (45.3% agree, 29.5% slightly agree).

Figure 4 shows respondents’ perceptions on acceptable amounts and frequency of non-research payments from Pharma to physicians. 48.1% of respondents considered an interaction worth JPY 10,000 (USD 92) or below acceptable, and 57.8% believed that the annual amount should be less than JPY 100 thousand (USD 917) or below. Furthermore, 77.8% of the respondents considered one interaction every few months as acceptable.

#### 3.2.4. Attitude towards Various Professional FCOI

Figure 5 shows the respondents’ perceptions towards potential FCOI in various professions, including court judges, referees, politicians, physicians, and business professionals. Overall, a larger percentage of respondents considered it problematic for court judges, politicians, or referees to receive gifts and meals from lawyers (agree 92.6%, slightly agree 6.3%), lobbyists (agree 75.5%, slightly agree 20.2%), or players (agree 89.4%, slightly agree 4.3%), respectively, than for physicians to accept gifts from Pharma (agree 58.5%, slightly agree 23.3%).

### 3.3. Qualitative Findings of Open-Ended Responses

A thematic analysis of the survey’s open-ended question, responded to by 56 (58.3%) of the respondents, identified 4 themes: (1) perception towards the FCOI; (2) concerns about the respondent’s treatment; (3) reasons for physician–Pharma interactions; and (4) possible solutions from the respondent’s perspective.
Theme 1: Perception towards the FCOI

Most respondents agreed that physicians should not receive honoraria except for research purposes. However, even the receipt of honoraria for research purposes was conditioned on the premise that such awards would not interfere with physicians’ ability to deliver high-quality, patient-centred care.

“Never permit benefits except for research purposes. Pharmaceutical companies and persons involved in research should realize that pharmaceutical companies are responsible for people’s lives. We don’t want doctors to accept one penny of honorarium”.

Furthermore, concerning lecture fees, a few respondents appeared to consider that physicians deserve to receive payment to some extent due to the physicians’ efforts in preparing for the lecture.

“Physicians are busy and time-restricted, so I think it’s reasonable for them to receive it (a lecture fee). But I disagree with any other kind of benefit-sharing between them, because that may hinder optimal selection of the prescription”.

Most respondents wanted the financial transactions minimized, except for reasonable research payments. This is in line with the results from our quantitative analysis. For example, more than half of the respondents considered non-research benefits of less than JPY 100,000 (USD 917) per year appropriate.
Theme 2: Concerns about the respondent’s treatment

Many respondents expressed concerns about the influence of physician–Pharma interactions on treatment decisions and the treatment they receive. Respondents mostly wanted doctors to treat patients based on their authentic judgment and not be influenced by these interactions.

“Doctors are also human, and if they are incentivized, then they may have to make a judgment that is favored by a pharmaceutical company. I’ve heard of such things, and I think doctors should put their patients first, as cliché as it may sound. Patients trust their doctors and put their lives in doctor’s hands. I hope that not all doctors are in favor of Pharma”.

“My former doctor used to go to ‘study meetings’ a lot. To treat my advanced cancer, this doctor strongly recommended a medicine that he had just learned about in these study meetings. Although I trusted him to treat me, having learned about the financial relationships between physicians and pharmaceutical industry in this research, now I would have made different decisions about his recommendations”.

In alignment with our quantitative analysis results, many respondents expressed concern about the influence of physician–Pharma interactions on physicians’ prescription decisions in general. Additionally, several respondents voiced unease about their physicians and the treatments they received.
Theme 3: Reasons of physician–Pharma interactions

Many respondents concluded that a lack of physician ethical norms could cause FCOI between Pharma and healthcare sectors.

“I think it comes down to the ethics and morality of each physician”.

Besides physicians’ morality, two respondents raised the current government policy for science as an ongoing issue of physician–Pharma interactions. They indicated that the decline in the national budget for scientific research led to the financial dependence on Pharma.

“I think there is also a problem with the way the Japanese government funds basic research, alongside physician ethics and morality, if we want to ensure that (healthcare) is not controlled by pharmaceutical companies”.

In contrast to Themes 1 and 2, neither of these two potential reasons were clarified in the quantitative results.
Theme 4: Possible solutions from the patient perspective

The open-ended responses about possible solutions for FCOI comprised three main positions: improving public awareness of physician–Pharma interactions, strengthening legal or public FCOI regulation, and educating healthcare professionals on ethical norms.

Several respondents suggested possible solutions for the FCOI between Pharma and healthcare sectors. Improving public awareness of physician–Pharma interactions was the idea suggested most often.

Additionally, regulating the FCOI by professional associations or law was suggested by several respondents while also acknowledging the need for receipts for transport, meals, and honoraria relating to lectures and research. These comments also supported the quantitative results that more than three-quarters of the respondents agreed with some form of regulation.

“If [doctors] give a lecture, [doctors] may get transportation, food, and honorarium. It is generally an acceptable amount of money. If the basis on which pharmaceutical companies decide the concrete amount of the payments to physicians is unclear, the criteria should be decided by the professional association or academic society, taking into account the age and experience of the speaker”.

On the other hand, a few respondents insisted a regulatory regime for physician–Pharma interactions based on each physician’s morality rather than on legal or public regulation. This would further indicate the need for the education of physicians on the perspective of ethical norms.

## 4. Discussion

In this study, we investigated the awareness and perceptions of FCOI between Pharma and physicians and its influence on patients’ trust among members of a Japanese cancer patient advocacy group, demographically representative of the general Japanese cancer patient population [43]. In addition, data from the survey’s open-ended question with 56 respondents were analysed thematically, yielding the 4 following themes: (1) perception towards the FCOI; (2) concerns about the respondent’s treatment; (3) reasons for physician–Pharma interactions; and (4) possible solutions from the respondents’ perspective. We primarily found that most of the respondents were aware of physician–Pharma interactions, although the extent differed based on the nature of the interaction. We also found that respondents mainly considered these interactions influential on clinical practice and agreed to the need for further regulation of physician–Pharma interactions. The following discussion were made in the way the argument addresses each research question presented in the introduction. We have also organized the discussion section, referring to both the quantitative and qualitative analyses.

### 4.1. RQ1. Awareness of Physician–Pharma Interactions

Regarding the awareness on physician–Pharma interaction, although the awareness of the interactions, such as receipt of stationeries (64.2%), travel fees (35.8%), lecture fees (25.3%), and disclosure of payments from Pharma to the healthcare sector (10.5%), were within the scope of previous studies (55–76%, 17–33.7%, 20–46%, 7.3–18.8%, respectively) [12,22,24,38,44,45], the awareness of receiving drug samples (53.1%) was significantly lower than studies in the US (87–93.9%) [44,46]. In contrast, the awareness of physicians conducting research for Pharma (47.4%), receipt of meals (50.5%) and receipt of textbooks (45.3%) were higher than US and Lebanese studies (23–32%, 22–37%, 35–37%), respectively [38,41,44]. The underlying reasons for such differences among countries are not obvious. One reason may be that, in Japan, there have been many reports in recent years about research misconduct involving pharmaceutical companies and physicians [47]. In addition, stationaries and educational gifts, which could be present in clinicians’ offices, were more likely to be recognized than personal payments for lectures, travel, or accommodations, a trend which we observed previously [44,46,48,49].

One of the most interesting findings about the awareness was that despite ongoing disclosure efforts since 2013, only about 10% of respondents knew major Japanese Pharma disclosed payments to physicians. Even so, the proportion of respondents who were aware of such disclosures was within the range reported for previous studies conducted in the US (7.3–18.8%) [12,22,24,45]. Nevertheless, this study’s respondents may have had a more robust general interest in awareness of FCOI between Pharma and healthcare sectors. The qualitative study’s findings also revealed a lack of attentiveness on this issue, suggesting a need for proper countermeasure. Notably, during the study period, on 15 January 2019, the Money Database, which summarizes a payment of honorarium and donations from Pharma to healthcare sectors was made publicly available in Japan. Since then, several academic papers and media reports have documented the financial relationships among Pharma and various healthcare sectors [20,34,35,36,50,51,52]. This potentially contributed to a recent increase in public attention to the issue of FCOI in Japan and warrants a follow-up study to further evaluate this issue.

Our qualitative analysis of the responses to the open-ended questions did not identify any theme directly associated with this research question. However, the qualitative study showed that there were multiple respondents who became aware of physician–Pharma interactions for the first time through this survey, for example: “I was surprised at how much I didn’t know about such relationships (data not shown)”; and “After looking at the questionnaire this time, there are many things I do not know (data not shown)”. In this respect, we believe that our survey worked as an advocacy activity not only as an academic study.

### 4.2. RQ2. Impacts of Physician–Pharma Interactions on Trust in Physicians

Regarding this research question, the findings of quantitative analyses demonstrated that a majority of the respondents considered that the presence of FCOI with Pharma has negative influence on the trust in physicians as harming trust in physicians. As shown in the Themes 1 and 2, the qualitative analyses basically yielded the findings consistent with those gained with the quantitative analyses.

However, the outputs on FCOI relating to Pharma-sponsored research were conflicting between the qualitative and quantitative analyses. The qualitative analysis indicated that some respondents believed FCOI relating to research was acceptable when framed in the context of Pharma–sponsored research. However, our quantitative findings noted that a larger proportion of the respondents (25%) reported a decreased trust in physicians than those who reported an increased trust (13.6%) in physicians. Furthermore, 78.1% of the respondents indicated a loss of trust when physicians received the payments from Pharma for letting their patients participate in a clinical trial. This finding starkly contrasts with previous studies in other countries, where the receipt of payments for lectures or conducting Pharma-sponsored research was viewed as a positive symbol of good physicians by the public [23,53,54]. Given that clinical trials can have positive consequences, including improving management strategies of specific conditions or expanding the treatment options of patients participating in clinical trials, this apparent scepticism of clinical trials among the Japanese public warrants attention. Nonetheless, FCOI between Pharma and physicians may cause bias in reporting research findings [55,56] or medical scandals [57]. Thus, proper control of FCOI must accompany research involving Pharma, about which we should enhance awareness.

### 4.3. RQ3. Perception on Physician–Pharma Interactions

Regarding RQ3, the quantitative analyses showed that more than three-quarters of respondents perceived general FCOI with Pharma to be influential in physician prescription in a certain way. The observed value (75.9%) was the highest in several previous studies reported in other countries, ranging from 29% in Turkey to 75.6% in Canada [38,44,45,46,58,59]. Additionally, more respondents agreed that physician–Pharma interactions would increase healthcare costs (72.6%) more so than other studies (33–67.3%) [44]. These findings are consistent with the findings observed in Theme 1 in the qualitative analyses. A possible reason for this may relate to the fact that our study population mainly consisted of cancer patients or their caregivers. Essentially, this population is characterized by more significant health risks with few treatment alternatives outside of seeing a physician [60,61,62] and associated regular and long-term care [63], leading to greater attention on how their treatments could be biased by these relationships [60].

Aspects of Japanese culture likely also contribute to our findings. In particular, the Japanese medical profession has traditionally been held sacred, with physicians considered above reproach due to their ethos. Often more so than in other countries. Moreover, there exists a deep sense of shame and extreme disdain associated with improper behaviour [64,65]. Consequently, physicians with inappropriate financial entanglements may be seen as a kind of “defilement”, leading to distrust or disgust of such physicians by Japanese patients more strongly than among those in other countries.

In addition, our quantitative research found that more than 80% of respondents shared their perception towards gifts accepted by physicians from Pharma as problematic, suggesting they should be minimized. Furthermore, over half of respondents reported that non-research payments exceeding JPY 100,000 (USD 917) per year inappropriate. This finding was also paralleled to the response cited in Theme 1 in the qualitative analysis. However, compared with other fields, fewer people considered that accepting gifts in the medical field as more problematic than the legal, athletic, and political fields and less problematic than the general business world. While this trend was similar to that observed in the US [41], our study showed that more respondents considered the physician–Pharma interactions to be unethical than their counterparts in the US (58.5% unethical and 22.3% somewhat unethical in our study, versus 44% unethical and 15% somewhat unethical in the US) [41]. Although we could not find an international comparison of physicians’ public perception related to ethical norms, our findings indicated that a greater percentage of the Japanese public might expect physicians to be ethical than the general public in other countries.

## 5. Clinical Implementation

In completing the questionnaires, respondents provided several key perspectives in their quantitative and open-ended qualitative responses to improve patient-centred care and manage FCOI between Pharma and healthcare sectors.

As our previous studies revealed, the current framework for regulating FCOI between Pharma and healthcare sectors in Japan still has room for improvement, particularly regarding transparency [20,21,35,36,66]. Similarly, in our study, both the quantitative and qualitative analyses demonstrated a consensus among the respondents that physicians, professional associations, and Pharma should exhibit higher ethical standards and self-regulate their FCOI rather than merely relying on government regulation. This perspective has important repercussions for future discussions on improving the transparency of FCOI in the Japanese healthcare system. One likely countermeasure would be developing voluntary disclosure databases initiated by organizations akin to Disclosure UK or Disclosure Australia [9,67]. Another option would include the rigorous and legally binding FCOI disclosure standards established by the Sunshine Act and the Open Payment database in the US or France’s Transparency in Healthcare database [5,10,11,12].

Second, our survey elucidated that nearly 20% of respondents were unaware of many kinds of physician–Pharma interactions despite representing a population with a high level of interest in medical issues. Given that numerous studies showed that FCOI between Pharma and healthcare sectors influence patient treatment [6,68,69] and that many patients wanted to know about these conflicts [12,54,70,71,72], it is noteworthy that patients could benefit significantly from knowledge about FCOI before choosing doctors and treatments [45,54,73]. Therefore, we suggest the possibility of exploring seminars focusing on direct communications with the general public to build stronger relationships between Pharma, healthcare sectors, and patients and to gain patient trust [74]. One study from Australia showed that small-sized workshops help improve patient knowledge about FCOI [75].

In the meantime, the potential for increasing patient awareness of FCOI between Pharma and healthcare sectors may disrupt the long-standing, trusting relationship between patients and their physicians. As Kanter previously hypothesized [73], our results suggest that patients with long-term and severe diseases such as cancer would lose faith in their physicians over FCOI. Moreover, these concerns extended to a physician’s judgment of treatment options, wherein some respondents expressed similar concerns about the treatment they received. In particular, the presentation of a medical professional’s FCOI in selecting treatment can be a very sensitive agenda. Specifically, this raises concerns about the general awareness of FCOI between Pharma and healthcare sectors, which could lead to some patients declining effective treatments suggested by their physicians. Therefore, careful consideration should be paid in communicating with patients about FCOI between Pharma and healthcare sectors.

Third, some respondents suggested a need to educate physicians about FCOI. According to *Medical Professionalism in the New Millennium: A Physician Charter*, maintaining trust by managing FCOI is one of the fundamental responsibilities for all physicians [76]. However, only one-third of Japanese medical schools undergo formal training on FCOI between Pharma and healthcare sectors [77]. Most physician–Pharma interactions are rooted in the first year of a physician’s training, after which physicians are continuously exposed to such interactions [78,79]. For this reason, few physicians question these interactions or their influence on their clinical practice, and some physicians are not forthright with patients about FCOI [77,80,81,82,83]. For example, more than 98% of Japanese medical students have financial interactions with Pharma [84]. Additionally, several studies revealed that junior physicians, more so than senior physicians, were more likely to accept interactions with Pharma [85,86] and consider them appropriate and valuable [86]. Therefore, we would suggest that all medical schools establish a curriculum on FCOI addressing these relationships’ undue influence on clinical practice [6,69,87] and their impact on patient trust and care [73]. Given that even preclinical medical students can interact with Pharma, it might be advisable to implement this curriculum early, by the second or third year in Japan [77,88,89].

## 6. Limitations

Our study has several limitations. First, our study sample was small, resulting from a low response rate of 18.3%. Indeed, the findings of the regression analyses about the awareness on physician–Pharma interaction and impact of such an interaction on trust in physicians could not identify any factors associated with these outcomes. This may be attributed to the large number of items prepared for this survey and the sensitive nature of the topic, which may have caused candidates to evade answering questions. On the flip side, it may be that only candidates with a relatively strong interest in or problematic awareness of FCOI between Pharma and healthcare professionals responded to this survey, limiting a generalizability of the study. Future studies should obtain responses from cancer patients and their caregivers more generally and employ approaches such as financial incentives to increase response rates. Second, our study was conducted among the members of a cancer patient advocacy organization, many of whom have much more contact with Pharma and healthcare professionals than the general public and presumably greater interest in medical issues. Furthermore, compared to Japan’s general cancer population in 2015, the current study population tended to be more male but with a similar age distribution [42]. Therefore, this study should be interpreted with caution about whether the results reflect the general Japanese public’s attitudes. Nevertheless, this study’s findings mirror existing perceptions among cancer patients and their caregivers towards the FCOI of physicians. Third, the current survey responses consisted of patients and non-patients. Although the analysis did not show statistical differences between the two groups, views on the physician–industry relationship may differ, and the number of analyses may have been insufficient to characterize differences. Fourth, we did not collect data on the physicians who oversaw the respondents’ treatments as cancer patients. Other unexamined confounding factors, such as physician specialty, details of their FCOI with Pharma, would influence the differences in respondent awareness and perception of physician–Pharma interactions. [12,24,59] Finally, although we have referred to prior surveys, the validity and reliability of the questionnaire used in this study have not been established. Despite these limitations, to the best of our knowledge, this is the first study to examine the awareness and perceptions of physician–Pharma interactions among non-healthcare professionals in Japan. The opinions of cancer patients and the supporting public could help explore appropriate FCOI management methods in Japan and other countries.

## 7. Conclusions

In this study, we found that a majority of the respondents were aware of some FCOI between healthcare sectors and Pharma in Japan, though the extent of this awareness differed depending on experience. Respondents also reported a significantly decreased trust in physicians who received personal gifts and payments versus those accepting only office-use-related gifts. In addition, several respondents expected that physicians should be highly ethical, minimize FCOI outside of research, and assume that FCOI disproportionately influences a physician’s clinical practice, increases healthcare costs, and lowers patients’ trust in physicians. Further steps are required to improve patient awareness of FCOI, patient trust, and transparency in healthcare. These include improving public awareness with seminars focusing on direct physician–patient communication, more patient-oriented regulation of FCOI, and educating medical students and physicians about FCOI.

## Figures and Tables

**Figure 1 ijerph-19-03478-f001:**
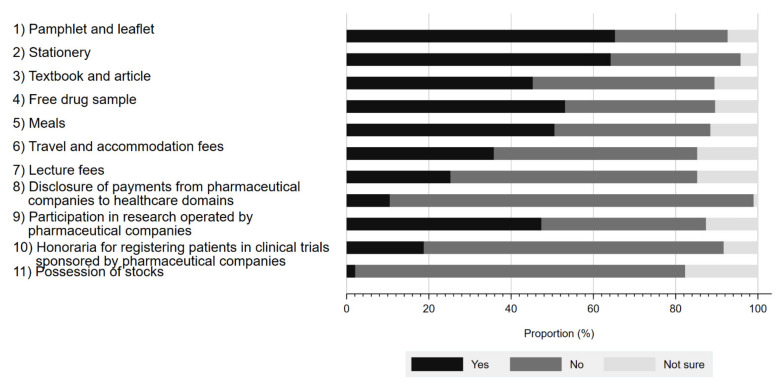
Respondents’ awareness of interactions between physicians and pharmaceutical companies.

**Figure 2 ijerph-19-03478-f002:**
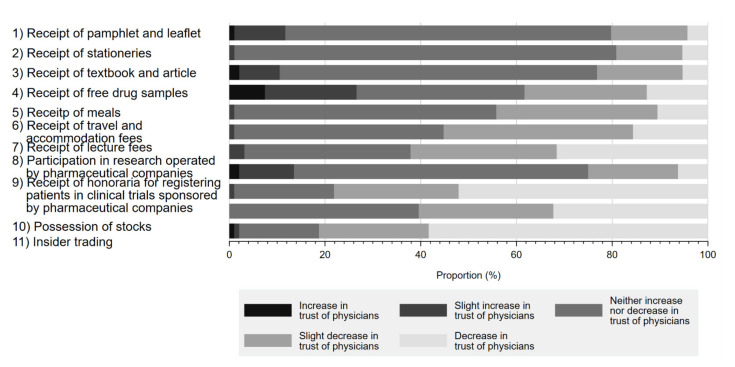
Influence of physician–Pharma interactions on trust in physicians.

**Figure 3 ijerph-19-03478-f003:**
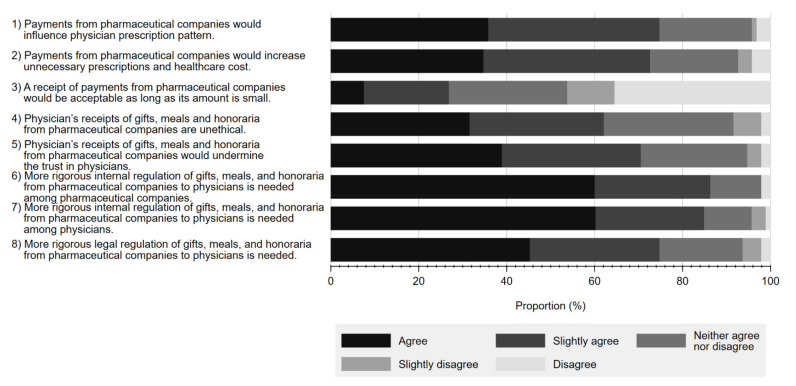
Respondents’ perceptions on statements regarding interactions between physicians and pharmaceutical companies and their associated regulations.

**Figure 4 ijerph-19-03478-f004:**
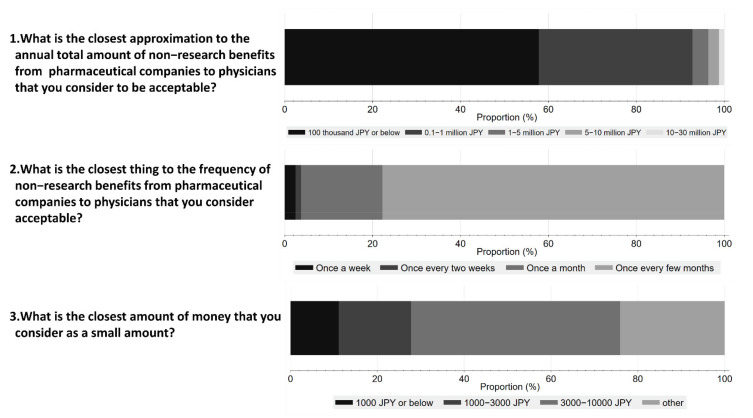
Respondents’ perception of acceptable amounts and frequency of non-research payments from pharmaceutical companies to physicians.

**Figure 5 ijerph-19-03478-f005:**
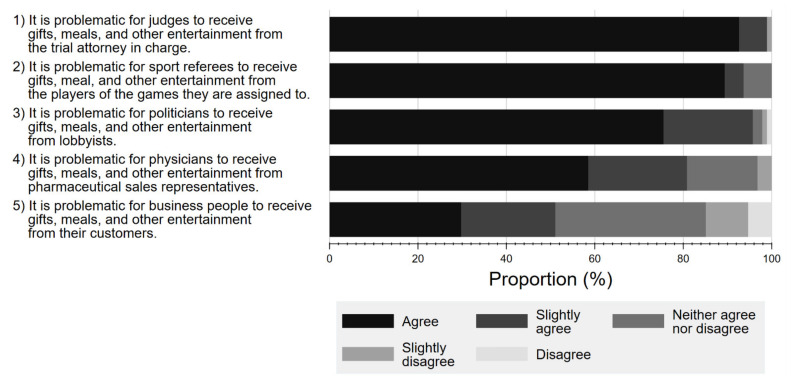
Respondent attitude about conflicts of interest among various professionals.

**Table 1 ijerph-19-03478-t001:** Demographic features of the study population.

Variables	
**Gender, N (%)**	
Male	63 (67.7)
Female	30 (32.3)
Missing	3
**Age category, N (%)**	
≦60	15 (16.0)
61–70	30 (31.9)
71–80	38 (40.4)
81–90	11 (11.7)
Missing	2
**Annual family income, N (%)**	
<JPY 2 million (<USD 18,349)	6 (7.0)
JPY 2–4 million (USD 18,349–36,697)	40 (46.5)
JPY 4–6 million (USD 36,697–55,046)	14 (16.3)
JPY 6–8 million (USD 55,046–73,394)	12 (14.0)
JPY 8–10 million (USD 73,394–91,743)	5 (5.8)
>JPY 10 million (>USD 91,743)	9 (10.5)
Missing	10
**Type of business, N (%)**	
Unemployed	52 (55.9)
Self-employed	15 (16.1)
Full-time job	11 (11.8)
Part-time job	7 (7.5)
Other	8 (8.6)
Missing	3
**Educational background, N (%)**	
Less than high school graduate	1 (1.1)
High school graduate	27 (29.0)
Associate degree or diploma	15 (16.1)
Bachelor’s degree or more (bachelor, master, and doctoral degree)	50 (53.8)
Missing	3
**Type of cancer, N (%)**	
Not cancer patient	20 (22.5)
Prostate cancer	24 (27.0)
Lung cancer	5 (5.6)
Breast cancer	5 (5.6)
Colorectal cancer	4 (4.5)
Gastric cancer	4 (4.5)
Other type of cancer	27 (30.3)
Missing	7
**Cancer’s stage, N (%) ***	
Stage 1	23 (35.9)
Stage 2	16 (25.0)
Stage 3	10 (15.6)
Stage 4	5 (7.7)
Unclear	10 (15.6)
Missing	5
**Year of diagnosis, N (%) ***	
2018	4 (5.8)
2017	6 (8.7)
2016	3 (4.4)
2015	4 (5.8)
Before 2015	52 (75.4)
**Hospital, N (%) ***	
Cancer hospital	20 (29.4)
National university	9 (13.2)
Private university	9 (13.2)
National municipal hospital	10 (14.7)
Private municipal hospital	11 (16.2)
Other hospital	9 (13.2)
Missing	1
**Previous cancer recurrence ***	
Yes	13 (20.0)
No	47 (72.3)
Not clear	5 (7.7)
Missing	4
**Count of previous cancer reoccurrence, N (%)**	
1	6 (42.9)
2	2 (14.3)
3	1 (7.1)
6	1 (7.1)
7	1 (7.1)
10	1 (7.1)
Unknown	2 (14.3)
**Treatment which you have ever had, N (%) *^,^ ****	
Anticancer drug	22 (16.5)
Molecularly targeted drug	5 (3.8)
Hormone therapy	23 (17.3)
Radiation therapy	37 (27.8)
Surgery	39 (29.3)
Other	4 (3.0)
Never	3 (2.3)
**Treatment which you have now, N (%) *^,^ ****	
Anticancer drug	2 (2.9)
Molecularly targeted drug	2 (2.9)
Hormone therapy	10 (14.5)
Radiation therapy	2 (2.9)
Other	11 (15.9)
Not having	42 (60.9)

* The denominator was the number of people with any type of cancer. (*n* = 69). ** Multiple answers were possible. Abbreviations: JPY—Japanese yen; USD—US dollar.

**Table 2 ijerph-19-03478-t002:** Demographic breakdown of respondents’ awareness of at least one physician–Pharma interaction.

Variables	Number (%)		*p*-Value
	Aware	Unaware or Not Sure	
**Gender**			
**Male**	49 (77.8)	14 (22.2)	0.053
**Female**	28 (93.3)	2 (6.7)	
**Age category**			
**≤60**	13 (86.7)	2 (13.3)	0.879
**61–70**	24 (80.0)	6 (20.0)	
**≥71**	39 (79.6)	10 (20.4)	
**Income**			
**Lower income (<JPY 4 million (<USD 36,697))**	39 (84.8)	7 (15.2)	0.129
**Higher income (≥JPY 4 million (≥USD 36,697))**	29 (72.5)	11 (27.5)	
**Job**			
**Employed**	30 (81.1)	7 (18.9)	0.576
**Unemployed**	45 (80.4)	11 (19.6)	
**Education**			
**High school graduate or less**	21 (75.0)	7 (25.0)	0.264
**Associate degree or more**	54 (83.1)	11 (16.9)	
**Cancer**			
**Non-cancer respondents**	17 (85.0)	3 (15.0)	0.433
**Cancer patients**	55 (79.7)	14 (20.3)	
**Cancer stage ^1^**			
**1**	17 (73.9)	6 (26.1)	0.287
**2–4**	26 (83.9)	5 (16.1)	
**Year ^1^**			
**2015–2018**	15 (88.2)	2 (11.8)	0.320
**Before 2015**	41 (78.9)	11 (21.1)	
**Hospital ^1^**			
**Other hospitals**	24 (80.0)	6 (20.0)	0.555
**Cancer special hospitals**	31 (81.6)	7 (18.4)	
**Previous cancer recurrence ^1^**			
**No or other**	41 (78.9)	11 (21.1)	0.246
**Yes**	12 (92.3)	1 (7.7)	
**Experience with pharmacotherapy ^1^**			
**No**	19 (73.1)	7 (26.9)	0.145
**Yes**	33 (86.8)	5 (13.2)	
**Experience with radiotherapy ^1^**			
**No**	23 (85.2)	4 (14.8)	0.447
**Yes**	29 (80.6)	7 (19.4)	
**Previous surgical treatment ^1^**			
**No**	22 (75.9)	7 (25.1)	0.292
**Yes**	32 (84.2)	6 (15.8)	

^1^ The analysis included only respondents with cancer. Abbreviations: JPY—Japanese yen; USD—US dollar; Pharma—pharmaceutical company.

**Table 3 ijerph-19-03478-t003:** Logistic regression analysis of respondents’ awareness of physician–Pharma interactions.

Variables	Odds Ratio (95% Confidence Interval)	*p*-Value
**Gender**		
**Male**	Ref.	
**Female**	4.00 (0.85–18.90)	0.080
**Age category**		
**≤60**	Ref.	
**61–70**	0.62 (0.11–3.49)	0.584
**≥71**	0.60 (0.12–3.10)	0.542
**Income**		
**Lower income (<JPY 4 million)**	Ref.	
**Higher income (≥JPY 4 million)**	0.47 (0.16–1.37)	0.168
**Job**		
**Employed**	Ref.	
**Unemployed**	0.95 (0.33–2.74)	0.931
**Education**		
**High school graduate or less**	Ref.	
**Associate degree or more**	1.64 (0.56–4.79)	0.369
**Cancer**		
**Non-cancer respondents**	Ref.	
**Cancer patients**	0.69 (0.18–2.70)	0.598
**Cancer stage ^1^**		
**1**	Ref.	
**2–4**	1.84 (0.48–6.97)	0.373
**Year ^1^**		
**2015–2018**	Ref.	
**Before 2015**	0.50 (0.10–2.51)	0.397
**Hospital ^1^**		
**Other hospitals**	Ref.	
**Cancer special hospitals**	1.11 (0.33–3.73)	0.869
**Previous cancer recurrence ^1^**		
**No or other**	Ref.	
**Yes**	3.22 (0.38–27.52)	0.286
**Experience with pharmacotherapy ^1^**		
**No**	Ref.	
**Yes**	2.43 (0.68–8.74)	0.173
**Experience with radiotherapy ^1^**		
**No**	Ref.	
**Yes**	0.72 (0.19–2.76)	0.633
**Previous surgical treatment ^1^**		
**No**	Ref.	
**Yes**	1.70 (0.50–5.74)	0.395

^1^ The analysis included only respondents with cancer. Abbreviations: JPY—Japanese yen, Pharma—pharmaceutical company, Ref—reference value.

**Table 4 ijerph-19-03478-t004:** Number and percent of respondents reporting decreased trust in at least one physician–Pharma interaction by respondent demographics.

Variables	Number (%)		*p*-Value
	Decrease Trust	Other	
**Gender**			
**Male**	55 (87.3)	8 (12.7)	0.157
**Female**	28 (96.6)	1 (3.4)	
**Age category**			
**≤60**	14 (93.3)	1 (6.7)	0.550
**61–70**	26 (89.7)	3 (10.3)	
**≥71**	44 (89.8)	5 (10.2)	
**Income**			
**Lower income (<JPY 4 million)**	41 (89.1)	5 (10.9)	0.590
**Higher income (≥JPY 4 million)**	36 (90.0)	4 (10.0)	
**Job**			
**Employed**	33 (98.2)	4 (10.8)	0.515
**Unemployed**	51 (91.1)	5 (8.9)	
**Education**			
**High school graduate or less**	23 (82.1)	5 (17.7)	0.089
**Associate degree or more**	61 (93.9)	4 (6.1)	
**Cancer**			
**Non-cancer respondents**	19 (95.0)	1 (5.0)	0.424
**Cancer patients**	62 (89.9)	7 (10.1)	
**Cancer stage ^1^**			
**1**	22 (95.7)	1 (4.3)	0.283
**2–4**	27 (87.1)	4 (12.9)	
**Year ^1^**			
**2015–2018**	14 (82.4)	3 (17.6)	0.316
**Before 2015**	46 (90.2)	5 (9.8)	
**Hospital ^1^**			
**Other hospitals**	26 (89.7)	3 (10.3)	0.517
**Cancer special hospitals**	33 (86.8)	5 (13.2)	
**Previous cancer recurrence ^1^**			
**No or other**	47 (90.4)	5 (9.6)	0.428
**Yes**	11 (84.6)	2 (15.4)	
**Experience with pharmacotherapy ^1^**			
**No**	24 (92.3)	2 (7.7)	0.398
**Yes**	33 (86.8)	5 (13.2)	
**Experience with radiotherapy ^1^**			
**No**	25 (92.6)	2 (7.4)	0.349
**Yes**	31 (86.1)	5 (13.9)	
**Experience with surgical treatment ^1^**			
**No**	24 (82.8)	5 (17.2)	0.119
**Yes**	36 (94.7)	2 (5.3)	

^1^ The analysis included only respondents with cancer. Abbreviations: JPY—Japanese yen; Pharma—pharmaceutical company.

**Table 5 ijerph-19-03478-t005:** Logistic regression analysis of the influence of physician–Pharma relationships on respondents’ trust.

Variables	Odds Ratio (95% Confidence Interval)	*p*-Value
**Gender**		
**Male**	Ref.	
**Female**	4.07 (0.48–34.21)	0.185
**Age category**		
**≤60**	Ref.	
**61–70**	0.62 (0.059–6.52)	0.713
**≥71**	0.63 (0.068–5.84)	0.683
**Income**		
**Lower income (<JPY 4 million)**	Ref.	
**Higher income (≥JPY 4 million)**	1.10 (0.27–4.40)	0.895
**Job**		
**Employed**	Ref.	
**Unemployed**	1.24 (0.31–4.94)	0.764
**Education**		
**High school graduate or less**	Ref.	
**Associate degree or more**	3.32 (0.82–13.44)	0.093
**Cancer**		
**Non-cancer respondents**	Ref.	
**Cancer patients**	0.47 (0.054–4.03)	0.488
**Cancer stage ^1^**		
**1**	Ref.	
**2–4**	0.31 (0.032–2.95)	0.306
**Year ^1^**		
**2015–2018**	Ref.	
**Before 2015**	1.97 (0.42–9.30)	0.376
**Hospital ^1^**		
**Other hospitals**	Ref.	
**Cancer special hospitals**	0.76 (0.17–3.48)	0.689
**Previous cancer recurrence ^1^**		
**No or other**	Ref.	
**Yes**	0.59 (0.10–3.42)	0.552
**Experience with pharmacotherapy ^1^**		
**No**	Ref.	
**Yes**	0.55 (0.098–3.08)	0.496
**Experience with radiotherapy ^1^**		
**No**	Ref.	
**Yes**	0.50 (0.089–2.78)	0.425
**Experience with surgical treatment ^1^**		
**No**	Ref.	
**Yes**	3.75 (0.67–20.93)	0.132

^1^ The analysis included only respondents with cancer. There were no statistically significant differences between the influence of physician–Pharmaceutical company relationships on respondents’ trust and each variable. Abbreviations: JPY—Japanese yen; Pharma—pharmaceutical company.

## Data Availability

The data presented in this study are available on request from the corresponding author. The data are not publicly available due to the privacy of participants.

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
