# Peer review of "Awareness and Perceptions among Members of a Japanese Cancer Patient Advocacy Group Concerning the Financial Relationships between the Pharmaceutical Industry and Physicians"

_ijerph, 2022, doi:10.3390/ijerph19063478_

Round 1

Reviewer 1 Report

Thank you for addressing this reviewers' remarks.

Author Response

Thank you for many of your previous helpful comments. We have revised the manuscript according to other reviewers and also amended the English language in the revised manuscript. 

I hope the revisions that we have made will meet your requests and that the revised paper is now publishable in the International Journal of Environmental Research and Public Health.

Reviewer 2 Report

  1. For the quantitative analysis in Sections 3.2.3 and 3.2.4, it is very simple because it only shows the percentage of the participants' perception. The questionnaire presents the demographic variable for which they cannot further analyse the difference or relationship or have an impact on participant awareness.
  2. In sections 3.2.1 and 3.2.2, the results of the analysis (p> 0.05) cannot appear in the supplement. Authors should set out the idea of research issues.
  3. The authors must indicate the research questions following the motivation of the research and the discussion must correspond to the research questions.
  4. In addition, the findings of the quantitative and qualitative analyses should be combined with the discussion.

Author Response

March 2, 2022

Dear Editor:

Thank you for your decision letter of February 6th, 2022, informing us a positive decision about our manuscript (ID- ijerph-1562140) entitled ‘Awareness and perceptions among members of a Japanese cancer patient advocacy group concerning the financial relationships between the pharmaceutical industry and physicians’ submitted to International Journal of Environmental Research and Public Health.

According to the comment and advice made by reviewer, I am herewith sending the revised manuscript and the point-by-point replies to the editor.

A point-by-point reply to the comments is indicated below. The changed parts are highlighted as yellow in the revised manuscript.

I hope the revisions that we have made will meet your requests and that the revised paper is now publishable in the International Journal of Environmental Research and Public Health.

Sincerely yours,

Anju Murayama

Medical Governance Research Institute

Residence Takanawa #201

2-12-13 Takanawa, Minato-ku, Tokyo 108-0074, Japan

P: +81-3-6455-7401

F: +81-3-6455-7505

E: [email protected]

Point by point comments for a reviewer

Comments 1:

For the quantitative analysis in Sections 3.2.3 and 3.2.4, it is very simple because it only shows the percentage of the participants' perception. The questionnaire presents the demographic variable for which they cannot further analyses the difference or relationship or have an impact on participant awareness.

Reply:

Thank you for your valuable comment. We agree with the reviewer's comment that our results for Sections 3.2.3 and 3.2.4 are very simple. When compared with Section 3.2.1, which investigated the awareness on physician-Pharma interaction, and Section 3.2.2, which investigated the impact of physician-Pharma interaction on trust in physicians, Sections 3.2.3 and 3.2.4 incorporated various kinds of items and it was difficult to set one factor to be narrowed down as the outcome variable to be explored in the regression analyses in these sections. In any case, the number of the respondents was very low in this study, and it was difficult to conduct the meaningful regression analyses for the Sections 3.2.1 and 3.2.2, and we could not find any associated factors in these sections. To address these concerns, we have revised the main text as follows, by adding further explanation in the Methods and limitation sections as follows.

(Line 186 to 190 in Page 4):

“When compared with Section 3.2.1, which investigated the awareness on physician-Pharma interaction, and Section 3.2.2, which investigated the impact of physician-Pharma interaction on trust in physicians, Sections 3.2.3 and 3.2.4 incorporated various kinds of items, and it was difficult to set one factor to be narrowed down as the outcome variable to be explored in the regression analyses in this section.”

(Line 585 to 594 in Page 19):

“Our study has several limitations. First, our study sample was small due to the low response rate of 18.3%. This may attribute to the large number of items prepared for this survey and the sensitive nature of the topic, which may have caused candidates to evade answering questions. On the flip side, it may be that only candidates with a relatively strong interest in or problematic awareness of FCOI between Pharma and healthcare professionals responded to this survey. Indeed, the findings of the regression analyses about the awareness on physician-Pharma interaction and impact of such an interaction on trust in physicians could not identify any factors associated with these outcomes. Future studies should obtain responses from cancer patients and their care-givers more generally and employ approaches to increase response rates, including monetary or non-monetary incentives.”

Comments 2:

In sections 3.2.1 and 3.2.2, the results of the analysis (p> 0.05) cannot appear in the supplement. Authors should set out the idea of research issues.

Reply:

(Page 8 to10)

Thank you for your helpful comment. We have moved the tables and results for Section 3.2.1 and 3.2.2 from the Supplementary Material 5 (Demographic breakdown of respondents’ awareness of at least one physician-Pharma interaction.) and Supplementary Material 6. (Logistic regression analysis of respondents' awareness of physician-Pharma interactions.) to the main text as Tables 2 and 3.

(Page 9 to 11)

We have also removed the table and results for Section 3.2.2 demonstrating the impact of physician-Pharma interaction and the findings of logistic regression models for the impact of physician-Pharma interaction from the Supplementary Material 6 (Number and percent of respondents reporting decreased trust in at least one physician-Pharma interaction by respondent demographics.) and Supplementary Material 7 (Table 5. Logistic regression analysis of the influence of physician-Pharma relationships on respondents' trust.) to the main text as Tables 4 and 5.

Comments 3:

The authors must indicate the research questions following the motivation of the research and the discussion must correspond to the research questions.

Response:

We have now added a new section clearly presenting the study aims and research questions following the introduction section.

(Line 99 to 111, Page 3)

“Study aims

                This study aimed to assess the awareness and perception of FCOI between physicians and Pharma among cancer patients and to explore its influence on patients' trust towards physicians and care in Japan. Also, we intended to discuss possible clinical implementation and improvements in the transparency of the FCOI between Pharma and healthcare sectors. We established the following three research questions to navigate the study.

  RQ1: How familiar are the cancer patients with the physician-Pharma interactions? (awareness)

 RQ2: By what factors do the physician-Pharma interactions influence patients' trust and care? (influence)

 RQ3: What are the cancer patients' perceptions towards the physicians-Pharma interactions? (perception)”

We have reorganized the whole discussion section in the way the argument there addresses each research question, using the subheadings of 4.1-4.3.

(Line 425 to 427, Page 16)

“The following discussion were made in the way the argument there addresses each research question presented in the introduction.”

Comments 4:

In addition, the findings of the quantitative and qualitative analyses should be combined with the discussion.

Responses:

Thank you for your valuable comment. We have carefully revised the discussion section and clinical implication section to reflect the findings of qualitative and quantitative analyses.

(Line 427 to 428, Page 16)

“We have also organized the discussion section, referring to both the quantitative and qualitative analyses.”

This manuscript is a resubmission of an earlier submission. The following is a list of the peer review reports and author responses from that submission.

Round 1

Reviewer 1 Report

The authors present cross-sectional survey results, with 18% participation, of Japanese individuals who are members of a cancer advocacy group. Most of the respondents were cancer patients. 

Abstract: include percentages behind qualitative language, for instance 'participants were generally neutral (XX%)', a 'large proportion (XX%) of interactions'

interested in why authors chose to include both patients and caregivers instead of focusing on one or the other, especially given a small percentage of survey respondents (18%) and smaller proportion of caregivers

Introduction: well-written, very interesting background. line 99: revise to specify that the respondents were both cancer patients and non-cancer patients who were involved in the advocacy group: 'among Japanese cancer patients and cancer advocates'

2.2 Data collection: did the authors consider a financial incentive to complete the survey to increase responses?

2.4: did authors assess gender as a predictor? This has been shown to be a significant predictor of physician-industry relationships and I wonder re: patient perception of relationships. 

3. Results - it is not clear how these questions were phrased for non-cancer patients. Were they asked how they were related to the cancer patient (eg, family member, caregiver) or if they were not even related to a cancer patient but just an advocate? I think this is relevant for result interpretation, especially when asking cancer-specific questions, like 'what kind of cancer do you have'. The authors should revise the text to refer to the "respondent' as opposed to 'the patient' since it is not clear if the respondent is the patient, family member, etc. In addition, were differences in these groups (patient vs non-patient) investigated? It would seem that this may be an important distinction based on the individual's interaction with the physician.

Figures: the font should be larger so easier to read. Please revise.

Discussion: lines 356-357, the authors should elaborate more on potential reasons why there may be cultural differences in either the perception or the industry relationships between Japan and the countries cited (US, Lebanon)

Lines 435. The authors could include discussion about the physician-patient relationship and disclosure on a personal level during treatment discussions 

Limitations: the authors should again address that the population is made up of both patients and their caregivers, and unless they have granular data to look at differences between these groups in responding, they should note that as a limitation

Author Response

December 6, 2021

ijerph-1454524

Dear Reviewer 1 and Reviewer 2,

              On behalf of my co-authors, I would like to express our gratitude for your interest in our manuscript entitled “Awareness and perceptions among members of a Japanese cancer patient advocacy group concerning the financial relationships between the pharmaceutical industry and physicians” and for the opportunity to submit a revised version.

             We hope you find the revised version suitable for publication in the International Journal of Environmental Research and Public Health. We look forward to hearing from you shortly.

Sincerely,

Anju Murayama

Medical Governance Research Institute,

2-12-13 Residence Takanawa 201, Takanawa, Minato-ku, Tokyo, Japan

Phone: +81-90-6321-6996

Awareness and perceptions among members of a Japanese cancer patient advocacy group concerning the financial relationships between the pharmaceutical industry and physicians

Response Letter

We would like to thank the Reviewers for their time and careful consideration of our manuscript. Please find below a detailed description of the revisions and our responses to the reviewers.

Reviewer 1

Comment

The authors present cross-sectional survey results, with 18% participation, of Japanese individuals who are members of a cancer advocacy group. Most of the respondents were cancer patients.

Abstract: include percentages behind qualitative language, for instance 'participants were generally neutral (XX%)', a 'large proportion (XX%) of interactions'

Reply

We appreciated your constructive comment on our study. Accordingly, we have revised the abstract as follows:

Page 1 Lines 31-34

Participants were generally neutral on how such interactions would affect physician trustworthiness (16.7%‒79.8% by the type of interactions). A majority of participants agreed that physician-Pharma interactions were unethical (62.1%), influenced physicians' prescribing behavior (74.7%), led to unnecessary prescriptions (72.6%), and negatively affected physician trustworthiness (70.5%).

Interested in why authors chose to include both patients and caregivers instead of focusing on one or the other, especially given a small percentage of survey respondents (18%) and smaller proportion of caregivers

Reply

We appreciate your important question. In this study, we focused on patient advocacy representatives lacking professional medical backgrounds as representatives of the general population. As you point out, it would be meaningful to focus on each group individually (patient versus caregiver), but this was not the main focus of this study, and the number of each group was small. Still, we analyzed patient and non-patient responses but found no significant differences in awareness of physician-Pharma interactions or change in physician trust associated with them. A more detailed examination of such differences that also accounts for different types of cancer remains an emphasis for a future study.

Introduction: well-written, very interesting background. line 99: revise to specify that the respondents were both cancer patients and non-cancer patients who were involved in the advocacy group: 'among Japanese cancer patients and cancer advocates'

Reply

We appreciate your comments. We have modified the manuscript as follows:

Page 3 Lines 97-99

In sum, the present study aimed to examine awareness of physician-Pharma interactions and their impact on physicians' trust while appraising perceptions of these payments among Japanese cancer patients and cancer patient advocates.

2.2 Data collection: did the authors consider a financial incentive to complete the survey to increase responses?

Reply

Thank you for this comment. It is indeed a possibility we would have had a better response rate had we considered financial incentives. However, we did not. Also, this study was conducted with the full cooperation of the cancer patient association without funding, making it difficult to incorporate financial incentives. In response to this comment, we have modified our limitations section to acknowledge the future use of financial incentives and other methods to increase the response rates.

Page 14 Lines 494-497

Additionally, while sociodemographic and clinical factors were not significantly related to the perception of physician-industry relationships, it remains possible that the number of respondents was insufficient. Future studies should obtain responses from cancer patients and their caregivers more generally and employ approaches like financial incentives to increase response rates.

2.4: did authors assess gender as a predictor? This has been shown to be a significant predictor of physician-industry relationships and I wonder patient perception of relationships.

Reply

We appreciate your interest. We considered that sociodemographic factors including gender, age, and income level might be associated with the respondents’ perception of physician-industry relationships. However, the number of female respondents was small, nor were gender differences the main topic of the study. Still, we performed secondary analyses on these factors and perceptions using logistic regression. The data shown in supplementary material 6 revealed no significant trends or differences.

Specifically, with respect to gender, females tended to consider financial physician-pharmaceutical company relationships on physician trust (Odds ratio 2.56, 95% CI 1.00-6.59, p >0.05, by univariable logistic analysis), but this was not significant. Perhaps due to the small sample size. Similarly, the multivariate logistic analysis did not find a significant difference in gender (Odds ratio, 2.12, 95% CI: 0.79 – 5.71, p>0.05).

We consider the relationship between social and clinical factors and the view of the physician-industry relationships as subjects for further study and modified our limitation section (Page 14 Lines 494-497) as described above in the previous comment.

  1. Results - it is not clear how these questions were phrased for non-cancer patients. Were they asked how they were related to the cancer patient (eg, family member, caregiver) or if they were not even related to a cancer patient but just an advocate? I think this is relevant for result interpretation, especially when asking cancer-specific questions, like 'what kind of cancer do you have'.

Reply

We appreciate your comment. To be clear, we only asked cancer patients about the type of cancer (e.g., colorectal cancer, gastric cancer, lung cancer, breast cancer, prostate cancer, and other cancer) clinical stage of their disease (e.g., stage and type of hospital they visited), type of hospital where the patient participant received treatment, the treatment participants had been receiving, treatments participants had ever received, and disease recurrence. As for the other questions, they were prepared by design to be answered by both cancer patients and non-cancer patients. We did not collect data on the relationship of non-cancer patients as caregivers, surviving family members, or friends, etc. We described in detail the process behind question preparation and included this as supplementary information. We also included the following sentence to emphasize the collection of data from both cancer patients and non-cancer patients alike.

Page 3 Line 110-111

Therefore, the target population of this study includes cancer patients and their caregivers.

The authors should revise the text to refer to the "respondent' as opposed to 'the patient' since it is not clear if the respondent is the patient, family member, etc.

Reply

Thank you for your comment. We have removed all use of the word “patient” where appropriate as it pertains to survey respondents and replaced “patient” with either "participant" or “respondent.”

In addition, were differences in these groups (patient vs non-patient) investigated? It would seem that this may be an important distinction based on the individual's interaction with the physician.

Reply

Thank you. We conducted analyses for differences between cancer patients and non-cancer patients. Neither the univariate nor multivariate analyses revealed significant group-related differences associated with the perception of the physician-industry relationships. We described these results in Supplementary Material 8. However, due to the small number of subjects for each group, the results should be interpreted with caution. To better focus on potential differences between patients and non-patients alike, a larger subject population is needed and remains a subject for future research. To emphasize these results and limitations, we have modified the wording of the result and limitation as follows.

Page 7 Lines 236-240

Our regression analysis revealed that none of the sociodemographic or clinical variables, including gender and the distinction between cancer patients and non-cancer patients, significantly predicted whether participants’ trust in physicians would decrease or not (p ≧ 0.05) (Supplementary Material 8).

Page 14 Lines 494-498

Additionally, while sociodemographic and clinical factors were not significantly related to the perception of physician-industry relationships, it remains possible that the number of respondents was insufficient. Future studies should obtain responses from cancer patients and their caregivers more generally and employ approaches like financial incentives to increase response rates.

Figures: the font should be larger so easier to read. Please revise.

Reply

We have amended the figures, following your suggestion.

Discussion: lines 356-357, the authors should elaborate more on potential reasons why there may be cultural differences in either the perception or the industry relationships between Japan and the countries cited (US, Lebanon)

Reply

Thank you. The reason for this is unclear but may be related in part to several recent in Japan in the past decade about research involving doctors and pharmaceutical companies. We have modified our discussion accordingly.

Pages 12 Lines 368-374

The underlying reasons for such differences among countries are not obvious. This may be because, in Japan, there have been many reports in recent years about research misconduct involving pharmaceutical companies and physicians [46], and conversely, cases of successful development. In addition, stationaries and educational gifts, which could be present in clinicians’ offices, were more likely to be recognized than personal payments for lectures, travel, or accommodations, a trend which we observed previously.[43, 45, 47, 48]

Lines 435. The authors could include discussion about the physician-patient relationship and disclosure on a personal level during treatment discussions

Reply

Thank you for this valuable perspective on this study. We need more transparent management of COI and patients need to know more about the FCOI of their healthcare providers. At the same time, patients should pay attention to such disclosures when making treatment choices. We have added additional text to include these discussions points.

Page 14 Lines 464-471

Moreover, these concerns extended to a physician’s judgment of treatment options, wherein some participants expressed similar concerns about the treatment they received. In particular, the presentation of a medical professional's FCOI in selecting treatment can be a very sensitive agenda. Specifically, this raises concerns about the general awareness of FCOI between Pharma and healthcare sectors, which could lead to some patients declining effective treatments suggested by their physicians. Therefore, careful consideration should be paid in communicating with patients about FCOI between Pharma and healthcare sectors.

Limitations: the authors should again address that the population is made up of both patients and their caregivers, and unless they have granular data to look at differences between these groups in responding, they should note that as a limitation.

Reply

Thank you for this valuable perspective on this study. We modified the limitation section accordingly.

Page 15 Line 506-509

Third, the current survey responses consisted of patients and non-patients. Although the analysis did not show statistical differences between the two groups, views on the physician-industry relationship may differ, and the number of analyses may have been insufficient to characterize differences.

Reviewer 2 Report

  1. The analysis of the questionnaire is only descriptive statistics. The authors should add some statistical methods in this paper. Otherwise, it is too simple.
  2. The validity and reliability of questionnaire cannot be found.
  3. The qualitative findings are from 56 participants. Hence, how to obtain theme 1 to theme 3, it should be stated such as coding, classification, and so on.

Author Response

December 6, 2021

ijerph-1454524

Dear Reviewer 1 and Reviewer 2,

              On behalf of my co-authors, I would like to express our gratitude for your interest in our manuscript entitled “Awareness and perceptions among members of a Japanese cancer patient advocacy group concerning the financial relationships between the pharmaceutical industry and physicians” and for the opportunity to submit a revised version.

             We hope you find the revised version suitable for publication in the International Journal of Environmental Research and Public Health. We look forward to hearing from you shortly.

Sincerely,

Anju Murayama

Medical Governance Research Institute,

2-12-13 Residence Takanawa 201, Takanawa, Minato-ku, Tokyo, Japan

Phone: +81-90-6321-6996

Response to Reviewer 2

Comment

The analysis of the questionnaire is only descriptive statistics. The authors should add some statistical methods in this paper. Otherwise, it is too simple.

Reply

We appreciate your comment. We performed logistic regression analysis on the sociodemographic variables collected with the survey. However, these assessments revealed no statistically relevant findings, as shown in the supplementary tables. Therefore, we elected to focus more on the descriptive statistics and participant responses. Despite the small overall number of subjects, we consider this study worth reporting because there have been no previous studies targeting non-medical people in Japan examining the perception of the relationships between medical professionals and pharmaceutical companies.

The validity and reliability of questionnaire cannot be found.

Reply

Thank you. We have referenced prior published work incorporating this and similar survey instruments in our citations. With the consent of the Journal, we could provide the instrument as a supplementary document, or if you like, make it available to you upon request. We have added the following sentence to the limitation section to acknowledge that the current survey has yet to be validated.

Page 15 Line 513-514

Finally, although we have referred to our prior surveys, the validity and reliability of the questionnaire used in this study have not been established.

The qualitative findings are from 56 participants. Hence, how to obtain theme 1 to theme 3, it should be stated such as coding, classification, and so on.

Reply

We followed Braun and Clarke’s method of thematic analysis: 1) familiarization, 2) generate initial codes, 3) search for themes, 4) review themes, and 5) define themes. These steps have been added to the methods section.

Pages 4 Lines 167-177

Lastly, participants’ responses to the open-ended questions on perceptions of FCOI were analyzed thematically, following Braun and Clarke.[41] This consisted of the following five steps: 1) familiarisation of the data, 2) generating initial codes, 3) searching for themes, 4) reviewing themes, and 5) defining themes. First, one member (AM) of our research team repeatedly read all the open-ended responses, identifying the units of meaning and generating codes to capture key thoughts and concepts contained in the responses. Next, this same team member categorized the codes thematically to identify themes. Verification of these themes, including their coherence and distinctiveness, was executed by the entire research team. This collaborative effort helped mitigate any unperceived biases associated with our individualized interests and prior research experience on financial and nonfinancial COI among healthcare professionals in Japan and the US.

Round 2

Reviewer 2 Report

The scale's validity and reliability should be established. It cannot be stated in the limitation. This is a very  big shortcoming. To publish in this journal, the authors must effort it to complete it.

Author Response

On behalf of my co-authors, I would like to express our deepest gratitude for your interest and constructive comments on our manuscript entitled “Awareness and perceptions among members of a Japanese cancer patient advocacy group concerning the financial relationships between the pharmaceutical industry and physicians.” Please find the attached Word file which we responded to your comments point by point and revised manuscript with tracke changes. 
